# Piezo2 expressed in proprioceptive neurons is essential for skeletal integrity

Eran Assaraf[1,2], Ronen Blecher[1,3,4], Lia Heinemann-Yerushalmi[1], Sharon Krief[1], Ron Carmel Vinestock [1],
Inbal E. Biton[5], Vlad Brumfeld[6], Ron Rotkopf [7], Erez Avisar[2], Gabriel Agar[2,8] & Elazar Zelzer[1,8 ✉]

In humans, mutations in the *PIEZO2* gene, which encodes for a mechanosensitive ion channel, were found to result in skeletal abnormalities including scoliosis and hip dysplasia. Here, we show in mice that loss of *Piezo2* expression in the proprioceptive system recapitulates several human skeletal abnormalities. While loss of *Piezo2* in chondrogenic or osteogenic lineages does not lead to human-like skeletal abnormalities, its loss in proprioceptive neurons leads to spine malalignment and hip dysplasia. To validate the non-autonomous role of proprioception in hip joint morphogenesis, we studied this process in mice mutant for proprioceptive system regulators *Runx3* or *Egr3*. Loss of *Runx3* in the peripheral nervous system, but not in skeletal lineages, leads to similar joint abnormalities, as does *Egr3* loss of function. These findings expand the range of known regulatory roles of the proprioception system on the skeleton and provide a central component of the underlying molecular mechanism, namely *Piezo2*.

[1] Department of Molecular Genetics, Weizmann Institute of Science, Rehovot 76100, Israel. [2] Department of Orthopedic Surgery, Assaf HaRofeh Medical Center, Sackler Faculty of Medicine, Tel Aviv University, Zerrifin 70300, Israel. [3] Department of Orthopedic Surgery, Assuta Ashdod University Hospital, Ashdod 7747629, Israel. [4] Ben Gurion University of the Negev, Beer-Sheva 8410501, Israel. [5] Department of Veterinary Resources, Weizmann Institute of Science, Rehovot 76100, Israel. [6] Department of Chemical Research Support, Weizmann Institute of Science, Rehovot 76100, Israel. [7] Bioinformatics Unit, Life Sciences Core Facilities, Weizmann Institute of Science, Rehovot 76100, Israel. [8] These authors contributed equally: Gabriel Agar, Elazar Zelzer. ✉email: eli.zelzer@weizmann.ac.il

Biomechanical signals regulate a large variety of biological processes at different levels of resolution, from the molecular to the system[1,2]. To understand the contribution of these signals, two main challenges need to be overcome. One is to identify the mechanism that converts the mechanical signal into biochemical signals, and the other is to determine the level at which the regulation takes place.

Piezo1 and 2 are calcium-permeable mechanosensitive ion channels. They are distinct from other ion channels by their large size and structure, which includes a series of four transmembrane helical bundles termed Piezo repeats, composing together flexible propeller blades[3–5]. A body of evidence accumulated in recent years highlights the importance of this family of ion channels in a variety of developmental and physiological processes. Piezo1 was found to be expressed in red blood cells (RBCs)[6], lungs[7,8], bladder[9], pancreas[10], and uterine endometrium[11]. Piezo1 plays a role in cell adhesion by maintaining integrin activation[12]. The ability of this channel to sense mechanical stimulation makes it sensitive to blood flow, suggesting its role in architecture of blood vessels in the lung during growth[13] and in regulation of vascular permeability in the lung[7]. In addition, Piezo1 takes part in urinary bladder extension compliance[9] and RBC hydration[6]. Piezo2 was found to be expressed in dorsal root ganglia (DRG) neurons[14], lung[15], gastrointestinal tract[16], and skin[17]. This channel is important for sensation of airway dilatation[8] as well as sensation of fine touch[17].

In addition, Piezo2 was shown to be expressed by proprioceptive mechanosensors, namely the muscle spindle and the Golgi tendon organ (GTO)[14]. These two organs share the ability to respond to changes in mechanical conditions, namely in muscle length (muscle spindles) or in actively generated force (GTOs). Specialized afferent fibers, termed proprioceptive neurons, transmit mechanical sensations from muscle spindle and GTOs via the DRG to the spinal cord[18–22]. Loss of Piezo2 in proprioceptive neurons results in severely uncoordinated body movements and abnormal limb positions, suggesting its requirement for the activity of these mechanosensors[14].

Another tissue where both Piezo1 and Piezo2 were found to be expressed is articular chondrocytes. It was shown that synergy between Piezo1 and Piezo2 channels provides high-strain mechanosensitivity to these cells, which might lead to cell death upon injury[23].

In humans, mutations in Piezo genes result in different pathologies. Mutation in the PIEZO1 gene leads to a rare hemolytic anemia named dehydrated hereditary stomatocytosis[24], and to a unique form of lymphatic dysplasia known as lymphatic dysplasia of Fotiou[25]. Mutations in the PIEZO2 gene have been held responsible for proprioception defects, scoliosis, and hip dysplasia[26] as well as arthrogryposis, a congenital contracture of multiple joints[27], perinatal respiratory distress[28], and muscle weakness[29].

The expression of Piezo genes in many body tissues and their involvement in various processes raise the question of whether the human skeletal phenotypes result from autonomous or nonautonomous effects. In this work, we study the involvement of Piezo2 in maintenance of skeletal integrity. We show that blocking the expression of Piezo2 in mouse chondrocytes or osteoblasts does not lead to alternations in skeletal morphology. Conversely, mice lacking Piezo2 in proprioceptive neurons acquire scoliosis and hip dysplasia, suggesting a nonautonomous role for Piezo2 in regulation of spine alignment and joint integrity. Targeting the expression of Runx3 and Egr3, which are necessary for the ontogenesis of the proprioceptive system, reinforces this notion, as it results in similar phenotypes. These findings demonstrate the significance of Piezo2 expression in the proprioceptive system for skeletal biology. Moreover, they expand the range of known regulatory roles of the proprioceptive system on the skeleton, advance the understating of the role of motion in pathologies of hip and spine and provide a mouse model for further studies of these diseases.

## Results

**Skeletal Piezo2 loss does not affect the spine or hip joint.** A recent report suggests that mutations in the human PIEZO2 gene result in a complex musculoskeletal phenotype involving scoliosis, hip dysplasia and hand deformities[30]. Piezo2 was shown to be expressed in numerous tissues; thus, relating the human phenotype to specific tissue expression is practically impossible. Because Piezo2 was previously shown to be expressed in skeletal cells such as chondrocytes[23], we sought to study its autonomous role in skeletal biology.

To study the possible developmental role of Piezo2 in mouse skeletogenesis, we focused on the limb. To block Piezo2 expression in limb mesenchyme lineages, we crossed Piezo2[f/f] mice with the Prx1-Cre mouse[31]. At P1, histological sections of proximal tibia of Prx1-Cre;Piezo2[f/f] cKO mice and control littermates stained with H&E and Safranin O were found to be similar (Supplementary Fig. 1A, B). Likewise, in situ hybridization for the marker genes Col1a1 (bone), Col2a1 (cartilage), Ihh (pre-hypertrophic chondrocytes), and Col10a1 (hypertrophic chondrocytes) showed no apparent differences between mutant and control limbs (Supplementary Fig. 1C). To expand our investigation, we analyzed micro-CT images and 3D reconstructions of tibia bones from P60 Prx1-Cre;Piezo2[f/f] cKO mice and control littermates (Supplementary Fig. 1D). No major effects on growth or morphology were observed in the mutant. Morphometric analysis of the bone images revealed reduced levels of bone mineral density in the Prx1-Cre;Piezo2[f/f] cKO mice relative to the control. No differences were found in the number and density of trabeculae, bone volume to total volume ratio or tissue mineral density (Supplementary Fig. 1E).

Next, given the human hip joint phenotype[30], we studied in detail the hips of Prx1-Cre;Piezo2[f/f] cKO mice. To quantify hip dysplasia, we used two radiographic measurements that are commonly used in the assessment of this condition in humans, namely the central edge angle (CEA)[32] and the acetabular index (Fig. 1a)[33,34] (see also Methods). Additionally, to study hip congruency we used the herein introduced congruency index, defined as the mean of multiple measurements of the distance along the joint line divided by the minimal value out of all these measurements (Fig. 1a and see also Methods). Using these indices, we analyzed micro-CT images and 3D reconstructions (Fig. 1b, c) as well as H&E-stained histological sections (Fig. 1d) of hip joints from Prx1-Cre;Piezo2[f/f] cKO mice and control littermates at P60. Results showed that loss of Piezo2 in the different lineages of limb mesenchyme did not affect hip joint shape or congruency.

To study the possible autonomous role of Piezo2 in spine alignment, we blocked the expression of Piezo2 in osteoblasts or chondrocytes. For that, Piezo2[f/f] mice were crossed with either Col1a1-Cre[35] or Col2a1-Cre[36] drivers, respectively. CT images of Col1a1-Cre;Piezo2 cKO and Col2a1-Cre;Piezo2 cKO spines were compared with images of control mice at P60 and P90. Coronal and sagittal views were used to evaluate scoliosis and kyphosis, respectively. Scoliosis was defined as a lateral curve of the spine (Cobb angle) greater than 10 degrees[37] and kyphosis was defined as excessive angulation of the spine in the sagittal plane compared to control animals. As seen in Fig. 2, comparison between cKO animals and control littermates showed that ablation of Piezo2 from cells expressing either collagen type I or II does not result in spinal deformity.

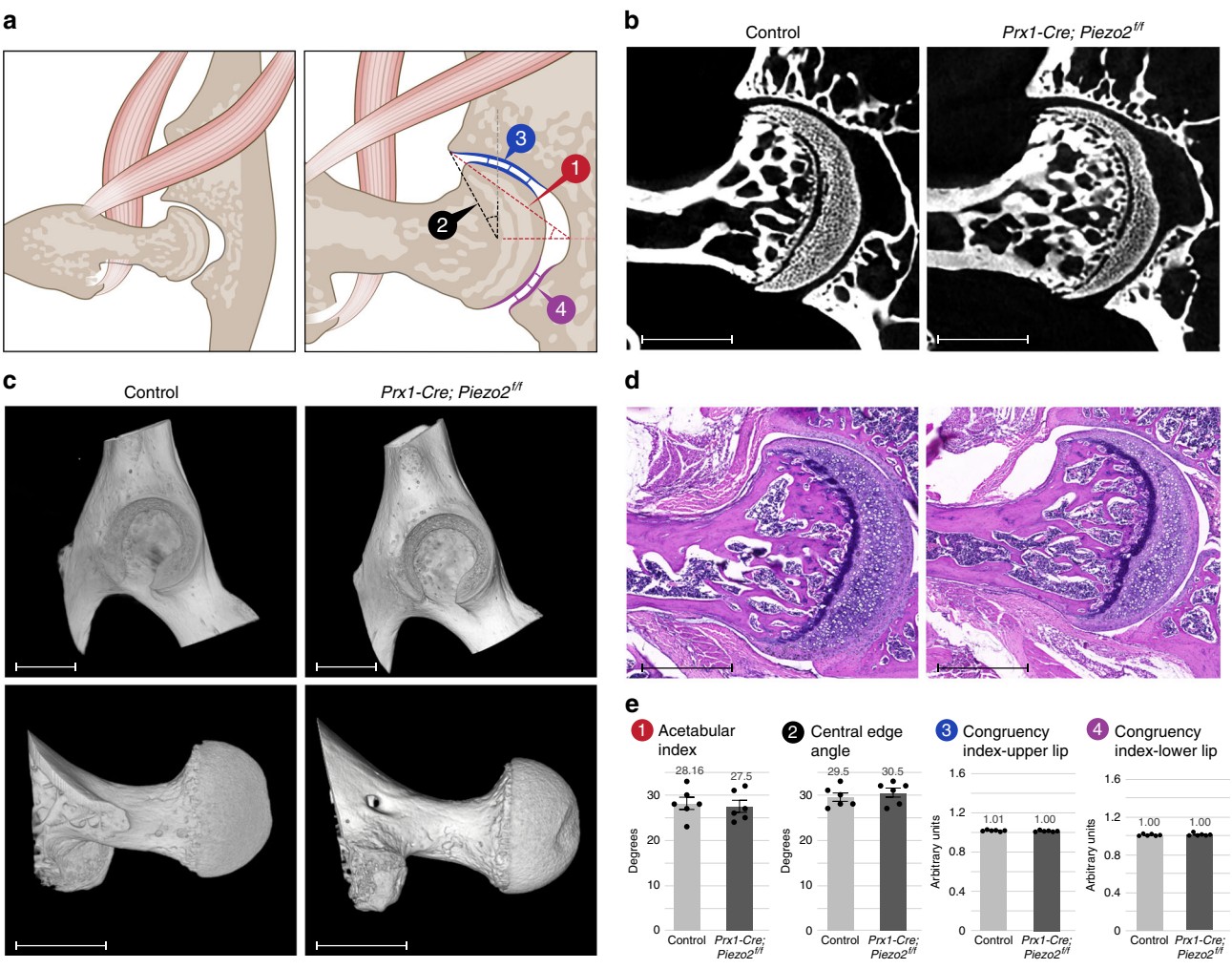

**Fig. 1 Loss of *Piezo2* in mesenchyme-derived tissues did not result in morphological changes of the hip joint. a** Illustrations of the hip joint showing the indices used to evaluate hip dysplasia. The red lines (1) mark the acetabular index, the black lines (2) mark the central edge angle, the blue (3), and purple (4) lines mark the zones of measurements for the congruency index in the upper and lower lip of the hip joint, respectively. **b** Ex vivo CT-scanned hip joints of representative P60 control (left) and *Prx1-Cre; Piezo2^f/f^* (right) mice (*n* = 6 in both groups). **c** 3D reconstruction of ex vivo CT scans of representative P60 control (left) and *Prx1-Cre; Piezo2^f/f^* (right) mice showing no signs of dysplasia of the acetabulum (top) or the femoral head (bottom). Data are from three independent experiments. **d** Histological H&E-stained sections of hip joints of representative P60 control (left) and *Prx1-Cre; Piezo2^f/f^* (right) mice showing no remarkable differences in the morphology of the hip. Data are from three independent experiments. **e** Graphs showing the measured values of the indices illustrated in **a** for P60 control (*n* = 6) and *Prx1-Cre; Piezo2^f/f^* (*n* = 6) mice. No significant differences were found in any of the measurements, as determined by Welch's two-sample *t*-test, ruling out the existence of hip dysplasia upon ablation of *Piezo2* in mesenchymal cells. *P*-values: 1, 0.73; 2, 0.47; 3, 0.32; 4, 0.9. Bar and whiskers represent mean value and SEM, respectively. Source data are provided as a Source Data file. Scale bars: 380 μm in **b**, 850 μm in (**c**, top), 770 μm in (**c**, bottom) and 510 μm in **d**.

We therefore concluded that although Piezo2 might be involved in regulating bone mineralization, it does not have an autonomous role in joint morphogenesis or spine alignment.

**Proprioceptive loss of *Piezo2* affects spine and hip joint.** The intact morphology of the spine and hip joint in mutants lacking *Piezo2* in skeletal cells prompted us to search for the tissue in which Piezo2 activity could affect the skeleton nonautonomously. Recently, we have demonstrated the regulatory role of the proprioceptive system in skeletal integrity[38,39]. Since Piezo2 is expressed and functions in proprioceptive neurons[14,40], we speculated that loss of *Piezo2* in these neurons could lead to skeletal abnormalities.

To block the activity of Piezo2 in the proprioceptive system, we crossed the *PValb-Cre* mouse with the floxed *Piezo2* mouse[14]. Spinal deformity was assessed by in vivo CT scans at three time points, namely P40, P60 and P90. At P40, while we did not observe a scoliotic phenotype in the *PValb-cre;Piezo2^f/f^* mice, we could already see an increment of the angulation of the spine in the sagittal plane compared to control littermates. By P60, some of the *Piezo2* cKO mice developed scoliosis in addition to the kyphosis (Fig. 3a).

Interestingly, while they were still present, the severity of both phenotypes was reduced at P90 (Fig. 3b).

Recently, we showed that *Runx3* KO mice, which lack proprioceptive circuitry and *Egr3* KO mice, in which muscle spindles fail to form whereas GTOs are retained, develop scoliosis. Therefore, we compared the patterns of spine malalignment between *PValb-Cre;Piezo2^f/f^* mouse and the above KO lines at P40, P60, and P90. In the sagittal plane, all three mutant lines displayed a similar pattern of thoracic malalignment that was characterized by a wide range of measurements, ranging from hypokyphosis to hyperkyphosis, as compared with

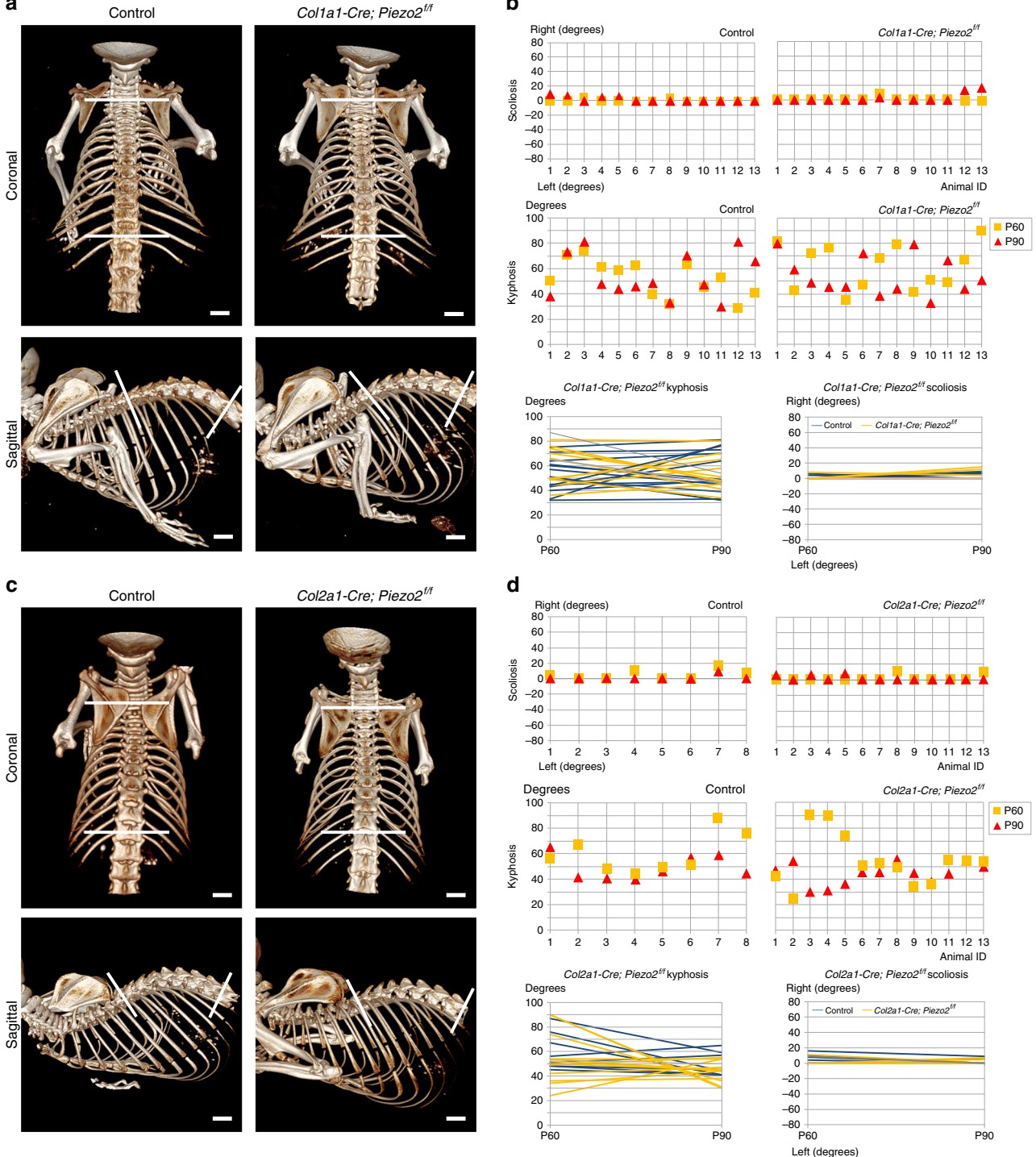

**Fig. 2 Loss of *Piezo2* in osteogenic and chondrogenic tissue did not affect spine alignment. a** In vivo CT-scanned skeletal images of representative control and *Col1a1-Cre; Piezo2^(f/f)* mice at P60 in coronal plane to evaluate scoliosis and sagittal plane to evaluate kyphosis. The two most tilted vertebrae on the caudal and rostral ends of the curve in both planes are indicated (white lines). The angle between the white lines parallel to the endplate of each of these vertebrae, termed the Cobb angle, was not affected by the osteogenic loss of *Piezo2* in both planes, as determined by *t*- and *F*-tests. **b** Graphs showing Cobb angle values for all control (left) and *Col1a1-Cre;Piezo2^(f/f)* mice (right, *n* = 13 in both groups) in the coronal (scoliosis) and sagittal (kyphosis) planes. The two line graphs at the bottom show the dynamics of Cobb angle for each measured mouse between P60 and P90. **c**, **d** Similarly, Cobb angles were not affected by the loss of *Piezo2* in chondrogenic lineages, as determined by *t*- and *F*-tests. Scale bars: 2.1 mm in **a**, 2 mm in (**c**, top left), 2.4 mm in (**c**, top right), 2.2 mm in (**c**, bottom right and left). Source data are provided as a Source Data file.

a narrow range of measurements in control littermates (Fig. 4a). Interestingly, as in *PValb-Cre;Piezo2^(f/f)* mice, the severity of thoracic malalignment in *Runx*3 KO and *Egr*3 KO mice was reduced at P90 (Fig. 4b, c). Levene's test for variance

homogeneity ruled out significant differences among the three groups of mutants at all time points, with *P*-values of 0.37 at P40, 0.60 at P60, and 0.11 at P90. In the coronal plane, all three lines displayed spinal malalignment. Interestingly, whereas *PValb-Cre*;

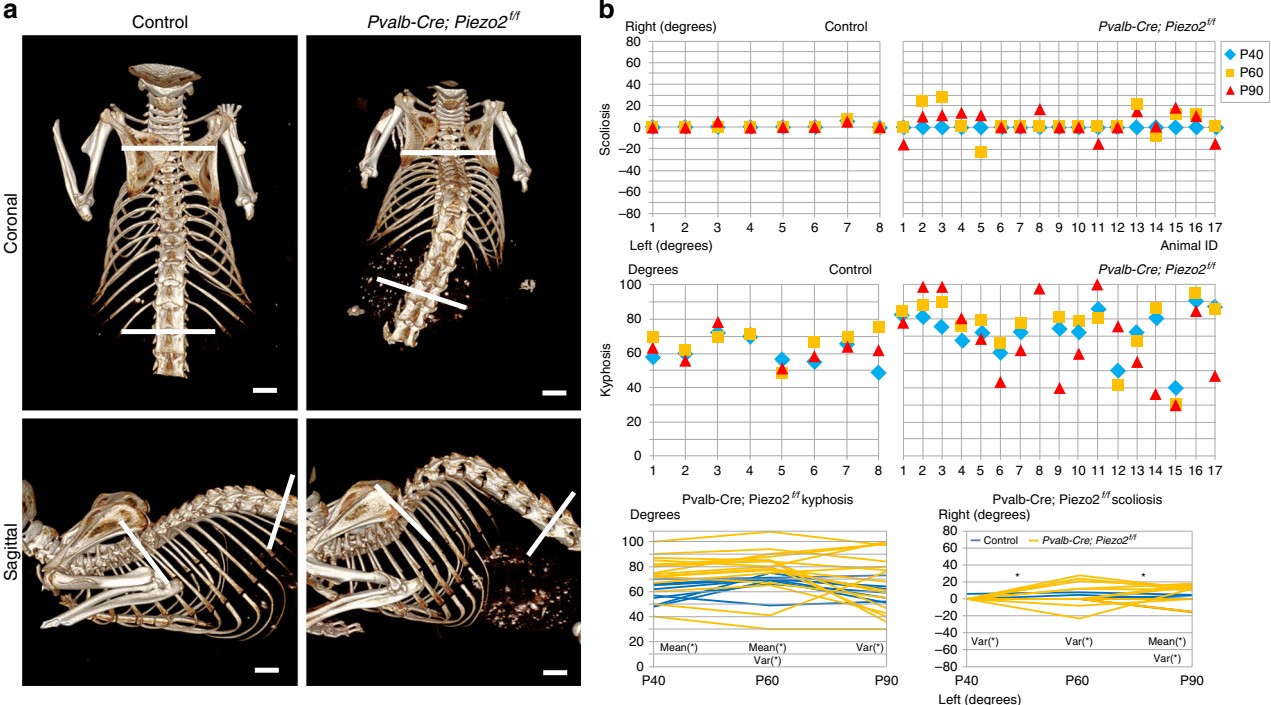

**Fig. 3 Loss of *Piezo2* in proprioceptive neurons results in spine malalignment. a** In vivo CT-scanned skeletal images of representative control and *Pvalb-Cre; Piezo2*$^{f/f}$ mice at P60, showing scoliosis (coronal plane) and kyphosis (sagittal plane) in the mutant compared to the control. **b** Graphs summarizing the Cobb angle values for all control (left, *n* = 8) and *Pvalb-Cre; Piezo2*$^{f/f}$ (*n* = 17) mice. Two graphs at the bottom show the dynamics of Cobb angle for each measured mouse between P40 and P90; significance of Mann–Whitney tests- or *F*-test is marked over the graphs by "mean(*)" or "var(*)", respectively. *P*-value for *Pvalb-Cre; Piezo2*$^{f/f}$ kyphosis P40 mw-0.17 f-infinity, P60 mw-0.64 f-0.0006, P90 mw-0.02 f-0.0003), *P*-value for *Pvalb-Cre; Piezo2*$^{f/f}$ scoliosis P40 mw-0.008 f-0.10, P60 mw-0.01 f-0.03, P90 mw-0.72 f-0.01). Source data are provided as a Source Data file. Scale bars: 2 mm in (**a**, top left), 2.1 mm in (**a**, top right), 2.5 mm in (**a**, bottom right and left).

*Piezo2*$^{f/f}$ spines exhibited a C-shaped scoliosis, i.e. one curvature, *Runx3* and *Egr3* KO mice exhibited an S-shaped, two-curvature scoliosis[38]. Moreover, mean cobb angle in *Runx3* KO mice was higher than in *PValb-Cre;Piezo2*$^{f/f}$ and *Egr3* KO mice at both P60 and P90 (Fig. 4a).

**Vertebrae and surrounding tissues are largely unaffected by *Piezo2* loss.** Having identified an association between loss of *Piezo2* in proprioceptive neurons and spine malalignment, we proceeded to assess the possible effects of this loss on the development and shape of skeletal elements and adjacent tissues, which could contribute to the phenotype. Examination of P10 histological spine sections from *PValb-Cre;Piezo2*$^{f/f}$ mice and control littermate stained with H&E or safranin O revealed no apparent differences in bone, cartilage or surrounding soft tissue. Similar results were obtained by in situ hybridization for marker genes for bone (*Col1a1*), cartilage (*Col2a1*), and tendons (*Scx*) (Supplementary Fig. 2). Next, we scanned by in vivo CT representative vertebrae (T4, T7, T10, T13, and L2) of P40 *PValb-Cre;Piezo2*$^{f/f}$ mice and control littermates. Reconstruction followed by registration enabled morphological comparison of the outer surfaces of matched mutant and control vertebrae. Morphometric analysis was performed using previously reported indices[41,42]. The ratios between anterior and posterior heights and right and left heights as well as between right and left superior facet angles were measured to assess the sagittal, coronal, and axial planes, respectively. Results showed that the mean index ratios at each plane and spinal level in *PValb-Cre;Piezo2*$^{f/f}$ mice were similar to those measured in control littermates (Fig. 5). These results suggest that aberrant development or gross morphology of vertebrae or surrounding

tissue are not likely to play a causative role in the spinal phenotype of *PValb-Cre;Piezo2*$^{f/f}$ mice.

**Piezo2 loss in proprioceptive neurons results in hip dysplasia.** Encouraged by the recapitulation of the human spinal phenotype, we studied the morphology of hip joints from P60 *PValb-Cre; Piezo2*$^{f/f}$ and control mice by micro-CT scans. We identified several abnormalities in the mutant, including elevated acetabular index and flattened type dysplasia with loss of joint congruency in both upper and lower sides (Fig. 6a). Moreover, the *Piezo2* cKO phenotype recapitulated a typical phenotype of adult hip deformity known as "cam"[43], a bony bulge at the head–neck junction that commonly results in impingement of soft tissue between the cam and acetabular pincer (Fig. 6b).

To better understand the temporal sequence of the development of the joint phenotypes, we analyzed H&E-stained sequential histological sections from P7-P60 mice. As seen in Fig. 6c, at P7 the mutant joint appeared similar to that of the control. However, by P14 signs of dysplasia were present, including excessive cartilaginous tissue at both side of the head–neck junction and incongruence of acetabular upper lip, suggesting that dysplasia starts to develop at the second postnatal week. At P45 and P60, a prominent femoral cam was seen at the greater trochanter side of the head–neck junction, and the acetabular joint line was found to be incongruent.

In humans, femoral acetabular impingement was suggested to be followed by hip osteoarthritis[44]. We therefore examined the affected hips for possible wear or inflammation. Safranin O-stained histological sections and in situ hybridization for the articular marker lubricin at P60 showed normal articular cartilage, with no signs of wear. Moreover, immunofluorescence

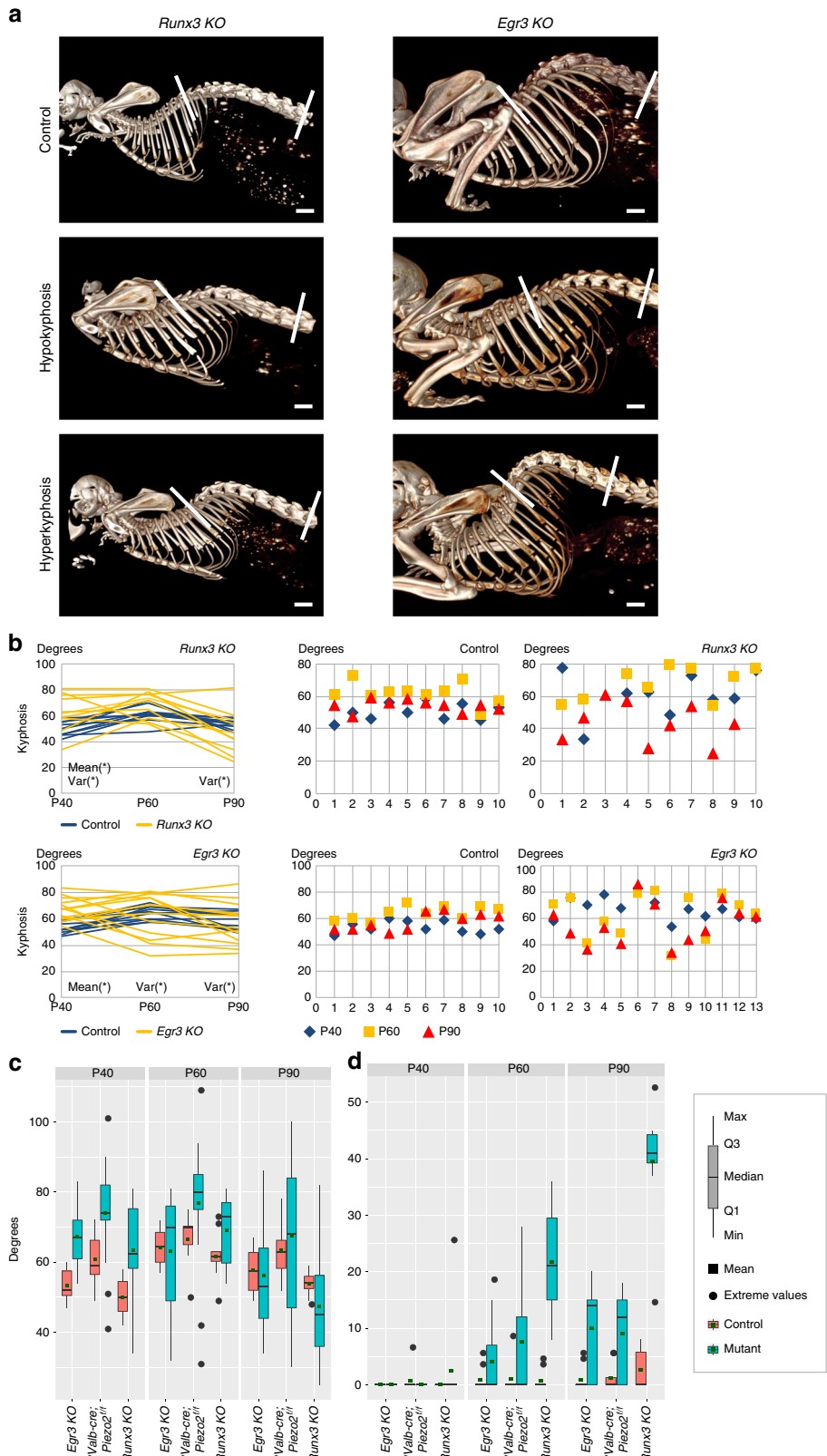

staining for the macrophage marker F4-80 ruled out an inflammatory process around the joint line (Supplementary Fig. 3).

**Runx3 deficiency recapitulates the hip dysplasia phenotype.** The observed spinal deformations are in line with our previous report that the proprioceptive system plays a major role in maintaining spine alignment[38]. Yet, the finding that loss of *Piezo2* in proprioceptive neurons led to abnormal hip morphology suggests a new function for the proprioceptive system in regulating hip morphogenesis. To further support this hypothesis, we analyzed *Runx3* KO mice, which lack proprioceptive

**Fig. 4 Comparative analysis of spine malalignment in *Runx3* KO, *Egr3* KO, and *PValb-cre;Piezo2^f/f* mice. a** In vivo CT-scanned skeletal images of representative control, *Runx3* KO and *Egr3* KO mice at P60 showing kyphosis in the KO mice but not in the control. **b** Graphs showing Cobb angles for all *Runx3* KO (top right, $n = 10$) and *Egr3* KO mice (bottom right, $n = 10$) and their controls (left, $n = 10$ for each group). Two graphs on the left show the dynamics of Cobb angle for each measured mouse from P40 through P60 to P90. As in *PValb-cre;Piezo2^f/f* mice, kyphosis was found to be progressive in most mutant animals between P40 and P60, with partial improvement at P90. Significance of t—or F-test is marked by "mean (*)" or "var (*)", respectively. P-values for *Runx3* KO: at P40, $p = 0.02$ (t-test) and $p = 0.006$ (F-test); at P60, $p = 0.07$ (t-test) and $p = 0.19$ (F-test); at P90, $p = 0.26$ (t-test) and $p = 0.0001$ (F-test). P-values for *Egr3* KO: at P40, $p = 0.00006$ and $p = 0.07$; at P60, $p = 0.83$ and $p = 0.001$; at P90, $p = 0.74$ and $p = 0.01$ (t—and F-test, respectively). Source data are provided as a Source Data file. **c, d** Box plots showing measurements of kyphosis (**c**) and scoliosis (**d**) in *PValb-Cre;Piezo2^f/f* ($n_{cKO} = 17$; $n_{control} = 8$), *Runx3* KO ($n_{KO} = 10$; $n_{control} = 10$), and *Egr3* KO mice ($n_{KO} = 13$; $n_{control} = 10$), at P40, P60, and P90. Scale bars: 2.4 mm in (**a**, top left), 1.7 mm in (**a**, middle left), 2 mm in (**a**, bottom left), 2 mm in (**a**, right; estimated based on measurements of animals of the same background and age).

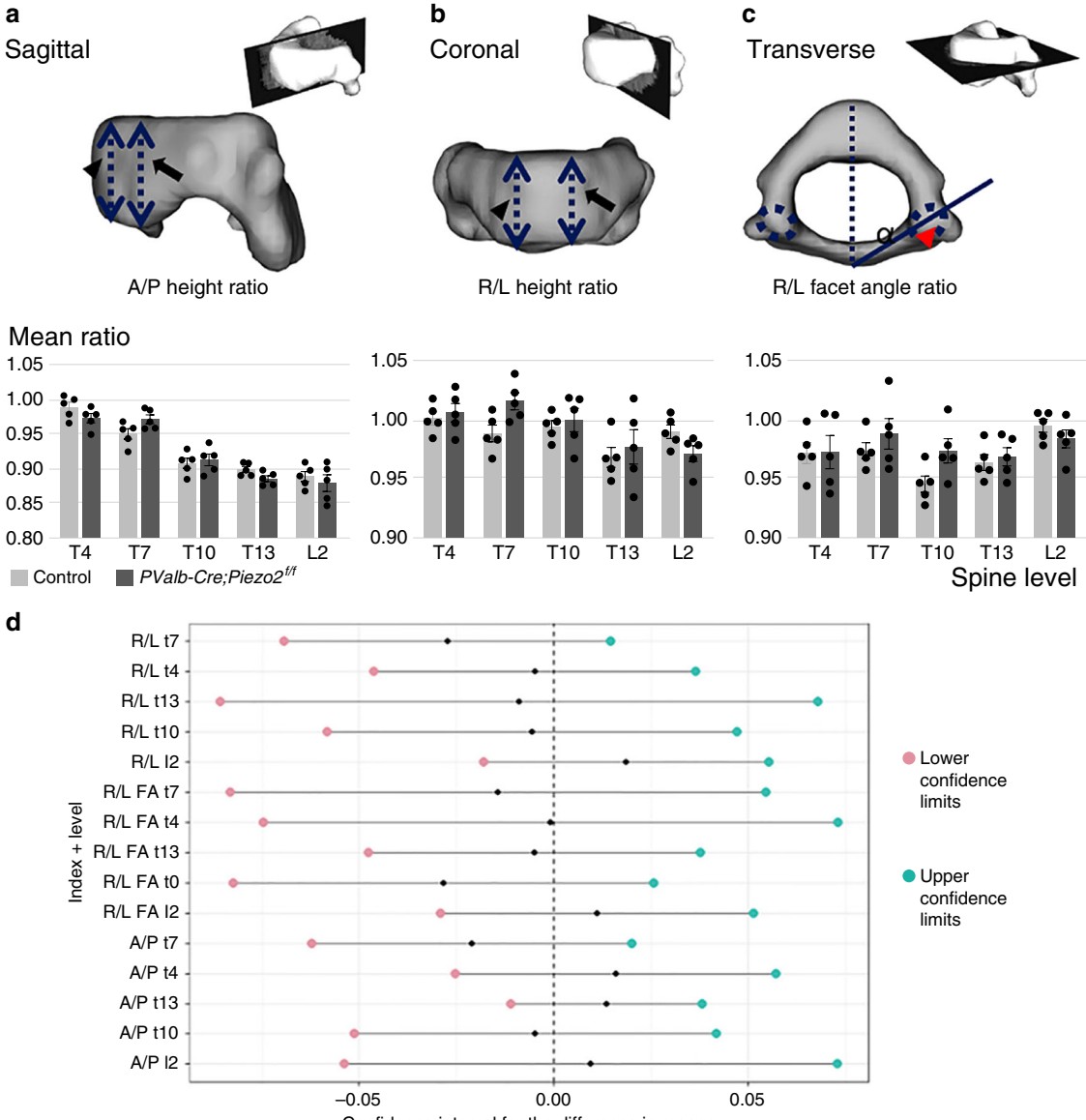

**Fig. 5 Vertebral shape is unaffected in *PValb-Cre;Piezo2^f/f* mice.** Reconstructed CT-scanned images of vertebrae illustrate the measured features (**a–c**): The ratios between anterior (dashed line indicated by arrowhead) and posterior (dashed line indicated by arrow) heights (**a**, sagittal plane, A/P ratio), between right (arrowhead) and left (arrow) heights (**b**, coronal plane, R/L ratio) and between right (dashed circle indicated by arrowhead) and left (dashed circle) superior facet angles (**c**, transverse plane, R/L facet angle ratio). Graphs below show the morphometric similarity between *PValb-Cre;Piezo2^f/f* (dark gray) and control (light gray) mice ($n = 5$ in each group). A/P anterior/posterior, R/L right/left. **d** Graph showing the 95% confidence intervals for differences in mean ratios. Red and green dots indicate upper and lower confidence limits, respectively, and black dots indicate the difference between control and mutant means ratio, which did not exceed the −0.1 to 0.1 range in all indices and at all levels. Data are presented as mean ± SEM; $n_{control} = 5$, $n_{cKO} = 5$. Source data are provided as a Source Data file.

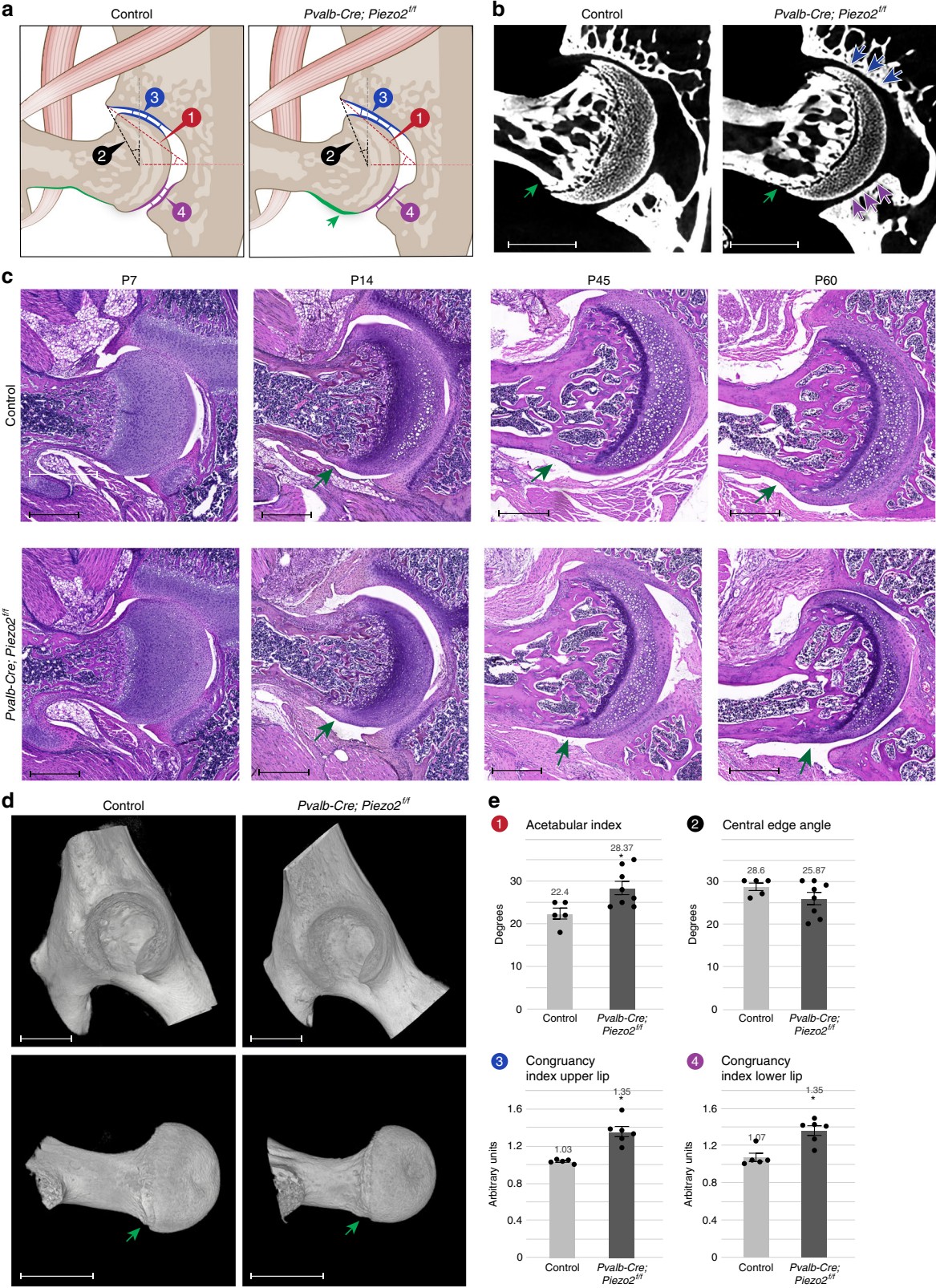

circuitry[45]. *Runx3* KO mice were evaluated at P60 by micro-CT imaging and histological sections and compared with control littermates. Results showed severe, irregular type hip dysplasia in all *Runx3* KO mice, which was manifested by a prominent cam over the femoral neck (Fig. 7b). Additionally, results showed significantly increased acetabular index representing a shallow acetabulum (Fig. 7c). Moreover, while in *PValb-Cre;Piezo2^{f/f}*

joints CEA values were similar to those of the control group (Fig. 7c), *Runx3* KO mice displayed significantly reduced CEA, indicating lateralized center of rotation, reduced joint stability and increased risk for dislocations. (Fig. 7c).

Since Runx3 is a transcription factor with a broad effect, we sought to substantiate a direct role of neural expression in the phenotype. For that, we ablated *Runx3* in the peripheral nervous

**Fig. 6 Loss of Piezo2 in proprioceptive neurons results in alternation of hip morphology. a** Illustrations of the hip joints of control (left, normal joint) and *Pvalb-Cre; Piezo2^{f/f}* (right, flattened type hip dysplasia) mice. Green arrow indicates femoral cam. **b** Ex vivo CT scans of P60 control (left, $n = 5$) and *Pvalb-Cre; Piezo2^{f/f}* mice (right, $n = 8$) hip joints. Flattening of the upper acetabular rim is seen in the mutant. Green arrows point at a femoral cam. **c** Histological sections at show first signs of femoral cam in a P14 *Pvalb-Cre; Piezo2^{f/f}* mouse with progression up to P60. Data are from three independent experiments. **d** 3D reconstruction of ex vivo CT scans show femoral cam (green arrow) in a P60 *Pvalb-Cre; Piezo2^{f/f}* mouse. Data are from three independent experiments. **e** Graphs showing increased acetabular index and congruency index (at both upper and lower lips) upon ablation of *Piezo2* in proprioceptive neurons. Control, $n = 5$ for all measurements; *Pvalb-Cre; Piezo2^{f/f}*, $n = 8$ for 1.2 and $n = 5$ for 3.4. Statistical significance as determined by Welch's two-sample *t*-test: 1, $p = 0.013$, 2, $p = 0.133$; 3, $p = 0.001$; 4, $p = 0.002$; asterisks indicate significant differences. Bar and whiskers represent mean value and SEM. Source data are provided as a Source Data file. Scale bars: 330 μm in (**b**), 590 μm in (**c**), 825 μm in (**d**, top), and 770 μm in (**d**, bottom).

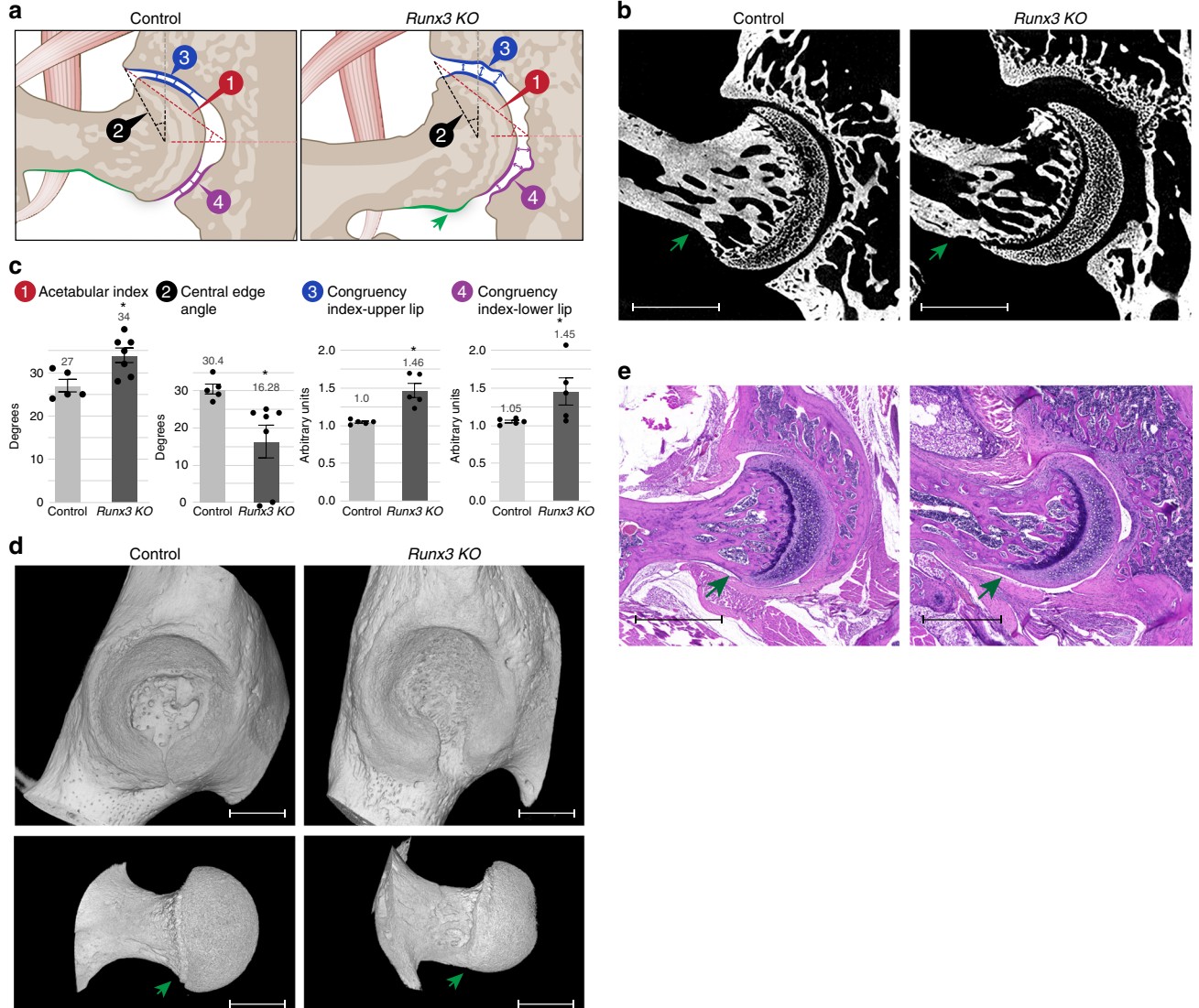

**Fig. 7 Total KO of *Runx3* results in severe hip dysplasia. a** Illustrations of the hip joint of control (left) and *Runx3* KO mice (right). The mutant exhibits irregular type hip dysplasia and femoral cam (green arrow). **b** Ex vivo CT scans of P60 control (left, $n = 5$) and *Runx3* KO (right, $n = 7$) mice showing severe irregular type hip dysplasia in the mutant. The green arrow points at prominent femoral cam. **c** Graphs indicating increased acetabular index, decreased mean CEA and hip incongruence over both upper and lower sides of the joint in the KO mice. Statistical significance as determined by Welch's two-sample *t*-test: 1, $p = 0.01$; 2, $p = 0.01$; 3, $p = 0.01$; 4, $p = 0.09$, $p = 0.031$ (Mann–Whitney tests, perfromed due to high variance of the sample), and $p = 0.0005$ (*F*-test); asterisks indicate significant differences. Bar and whiskers represent mean value and SEM. Source data are provided as a Source Data file. **d** 3D reconstruction of ex vivo CT scans at P60 show femoral cam (green arrow) and acetabular dysplasia in the *Runx3* KO mice. Data are from three independent experiments. (**e**) Histological H&E-stained sections through P60 hip joints show femoral cam (green arrow) and acetabular dysplasia in the *Runx3* KO mice. Data are from three independent experiments. Scale bars: 230 μm in (**b**), 815 μm in (**d**, top), 820 μm in (**d**, bottom), and 940 μm in **e**.

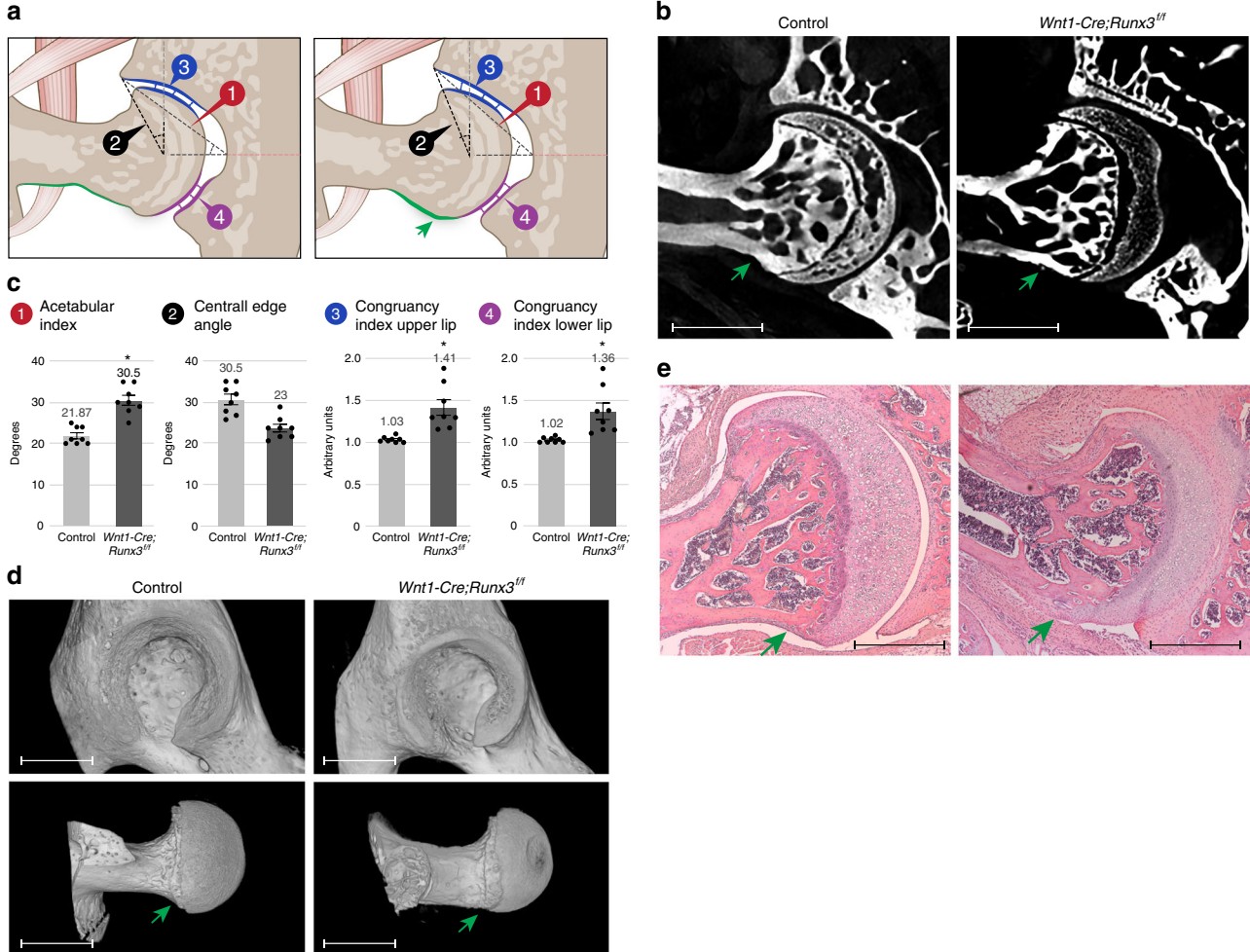

**Fig. 8 Loss of *Runx3* in neural tissue results in hip dysplasia. a** Illustrations of the hip joint of control (left) and *Wnt1-Cre;Runx3^f/f* mice (right). *Runx3* cKO mouse exhibits flattened type hip dysplasia and femoral cam (green arrow). **b** Ex vivo CT scans of P60 control and *Wnt1-Cre;Runx3^f/f* hip joints ($n = 8$ in both groups) show flattened upper acetabular rim and femoral cam in the mutant. **c** Graphs indicating increased acetabular index, decreased mean CEA and hip incongruence over both upper and lower sides of the joint in the cKO mice ($n = 8$ in both groups). Statistical significance as determined by Welch's two-sample *t*-test: (1) $p = 0.00006$; (2) $p = 0.0007$; (3) $p = 0.004$; (4) $p = 0.01$; asterisks indicate significant differences. Bar and whiskers represent mean value and SEM. Source data are provided as a Source Data file. **d** 3D reconstruction of ex vivo CT scans at P60 show femoral cam (green arrow) and acetabular dysplasia in the cKO mice. Data are from three independent experiments. **e** Histological H&E-stained sections through P60 hip joints show femoral cam (green arrow) in the cKO mice. Data are from three independent experiments. Scale bars: 365 μm in **b**, 815 μm in **d**, and 230 μm in **e**.

system using *Wnt1-Cre* driver[46] crossed with *Runx3^f/f* mice[45]. Micro-CT scans and histological sections of P60 *Wnt1-Cre; Runx3^f/f* mice and control littermates showed that ablation of *Runx3* from neural tissue resulted in flattened type hip dysplasia together with the typical cam phenotype, with extension of articular cartilage covering the cam over the femoral neck (Fig. 8b). Like in *Runx3* KO line, both CEA and acetabular index were significantly affected albeit to a lesser extent, and the congruency index at both sides of the acetabulum was markedly elevated (Fig. 8c).

In addition to its role in the proprioceptive system, Runx3 plays a role in differentiation of chondrocytes[47] and osteoblasts[48]. We therefore examined whether these functions of Runx3 were associated with the hip dysplasia phenotype. To answer this question, we ablated *Runx3* in osteoblasts or chondrocytes using *Col1a1-Cre* or *Col2a1-Cre* drivers, respectively, crossed with the *Runx3^f/f* mice[45]. Micro-CT scans of P60 *Col1a1-Cre;Runx3^f/f* mice or *Col2a1-Cre;Runx3^f/f* mice and control littermates showed that ablation of *Runx3* from osteoblasts or chondrocytes did not result in hip dysplasia, as no significant difference were measured in

CEA or acetabular index and joint congruency was found to be preserved. Moreover, the shape of the head–neck junction was regular, with no signs of a bony cam (Supplementary Fig. 4). These results suggest that *Runx3* expression in the skeleton is dispensable in the process of hip morphogenesis.

**Muscle spindle loss results in hip dysplasia**. Proprioceptive signaling is mediated by two types of peripheral sensors, the muscle spindle and the GTO. To determine the relative contribution of muscle spindles to the hip dysplasia phenotype, we examined mature mice deficient in *Egr3*, in which muscle spindles fail to survive whereas GTOs are retained[49]. Micro-CT scans and histological sections revealed that P60 *Egr3* KO mice developed a mild cam over the head–neck junction, without clear manifestations of dysplasia. However, the CEA was found to be significantly elevated, and congruency index confirmed mild joint incongruence (Fig. 9). These results reinforce the involvement of muscle proprioceptors in hip modeling. Moreover, the reduced phenotype severity as compared with *Runx3* mutants suggest that

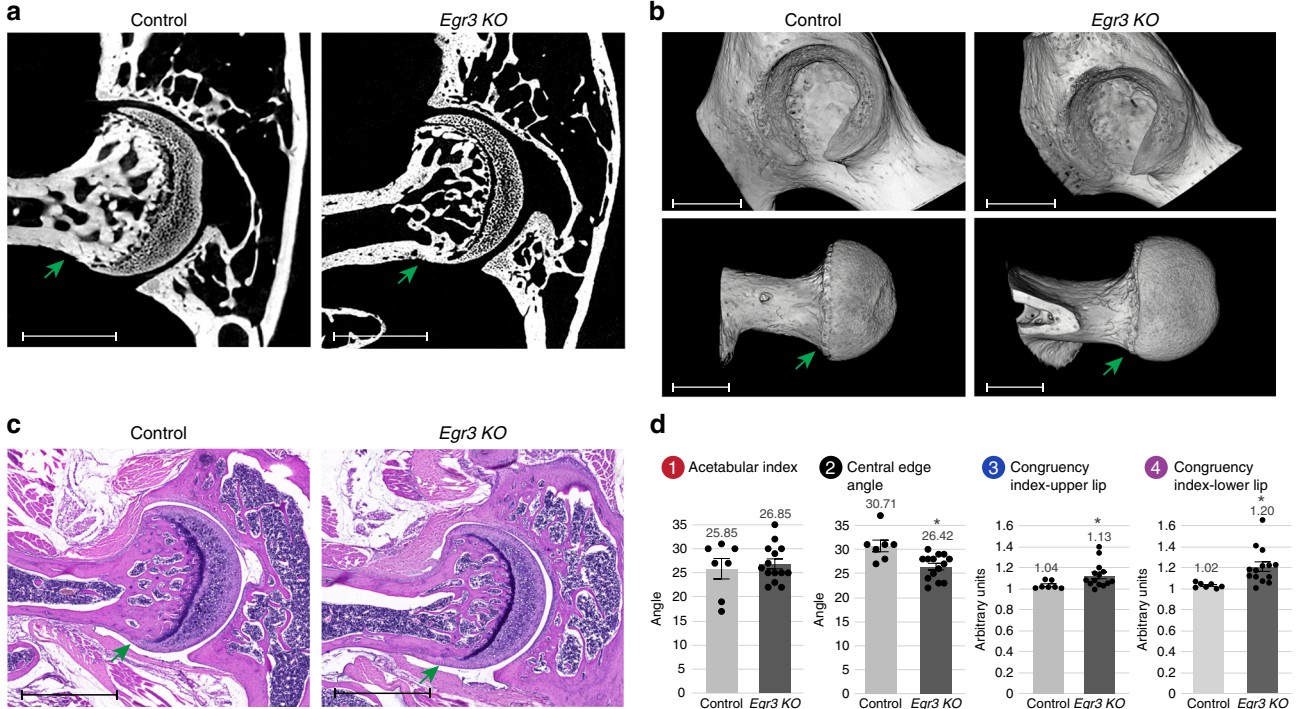

**Fig. 9 Loss of muscle spindle alone results in hip dysplasia of reduced severity. a** Ex vivo CT scans of P60 control (left, *n* = 7) and *Egr3* KO (right, *n* = 14) mice show flattening of upper acetabular rim and femoral cam (green arrow) in the mutant. **b** 3D reconstruction of ex vivo CT scans at P60 show femoral cam (green arrow) and acetabular dysplasia in the *Egr3* KO mice. Data are from three independent experiments.(**c**) Histological H&E-stained sections through P60 hip joints show femoral cam (green arrow) in the *Egr3* KO mice. Data are from three independent experiments. **d** Graphs indicating increased CEA, increased mean acetabular index and hip incongruence over both upper and lower sides of the joint in the *Egr3* KO mice ($n_{control}$ = 7; $n_{KO}$ = 14). Statistical significance as determined by Welch's two-sample *t*-test: (1) *p* = 0.68; (2) *p* = 0.001; (3) *p* = 0.014; (4) *p* = 0.002; asterisks indicate significant differences. Bar and whiskers represent mean value and SEM. Source data are provided as a Source Data file. Scale bars: 270 μm in (**a**), 825 μm in (**b**, top), 805 μm in (**b**, bottom), and 350 μm in **c**.

both receptor types are required for the formation of a normal hip joint.

## Discussion

In this work, we show that the expression of the mechanosensitive ion channel *Piezo2* in proprioceptive neurons is essential for both spine alignment and hip joint integrity. This finding provides molecular understanding of how the proprioceptive system regulates the skeleton. Utilizing mouse genetics, we show that loss of *Piezo2* in proprioceptive neurons, but not in chondrogenic or osteogenic lineages, led to spine malalignment and misshapen joints. Spinal malalignment of the *Piezo2* mutants provides further support to the regulatory role of the proprioceptive system in spine regulation. Finding similar joint abnormalities in mutants for the proprioceptive system regulators *Runx3* and *Egr3* confirms the role of this system in hip joint morphogenesis. Overall, this work expands our emerging concept[50] and firmly establishes the proprioceptive system as a central regulator of skeletal integrity and Piezo2 as a key component in this regulation (Fig. 10).

Over the years, numerous studies in different organisms, including chick[51], mouse, and zebrafish[52], have revealed that muscle-generated forces are necessary for joint development both in utero and postnatally[53–55]. In the absence of contracting muscles during embryogenesis, some joints fail to form, establishing the paradigm that movement is a necessary component in joint development[53,56,57]. However, this paradigm refers to two opposing situations, namely the existence or total absence of movement, disregarding everything in between. This dichotomous view completely prevents us from asking if there are

"right" and "wrong" types of movement, in other words, if the organism needs to move in certain ways for its joints to develop properly.

Previously, it has been reported that mutants for *Runx3* and *Egr3* and mice lacking *Piezo2* in the proprioceptive system exhibited severely impaired gait patterns[14,45,49]. Our finding of aberrant hip joint morphology in these mice correlates between impaired gait and aberrant joint morphology. This correlation suggests that specific patterns of movement are necessary for proper joint morphogenesis. It would be interesting to explore how these patterns of movement regulate joint morphogenesis. One possibility is that patterns of movement induce patterns of biomechanical signals that, in turn, are translated into specific molecular signals that regulate tissue morphogenesis.

Interestingly, we found that no other large joint except the hip was affected by the loss of *Piezo2* in proprioceptive neurons. Differential effect on joint development was previously reported in mutant embryos lacking functional muscles. In mice, for example, while the hip, elbow and shoulder joints were lost, the knee and the finger joints remained intact[53]. Explaining this variation is not easy. Yet, differences in the genetic developmental program between joints may be part of the explanation. For example, deletion of *Tgfbr2* in early limb mesenchyme results only in interphalangeal joint fusion[58,59], whereas in *Gdf5*-null mice, carpal, certain phalangeal and tarsals joints are abnormal[60]. Another possible explanation for the uniqueness of the hip joint is related to its anatomy. The pelvic bone is made of three different ossification centers, namely of the ilium, ischium and pubis. These three ossification centers merge into one at the center of the acetabulum[61]. It is possible that specific patterns of

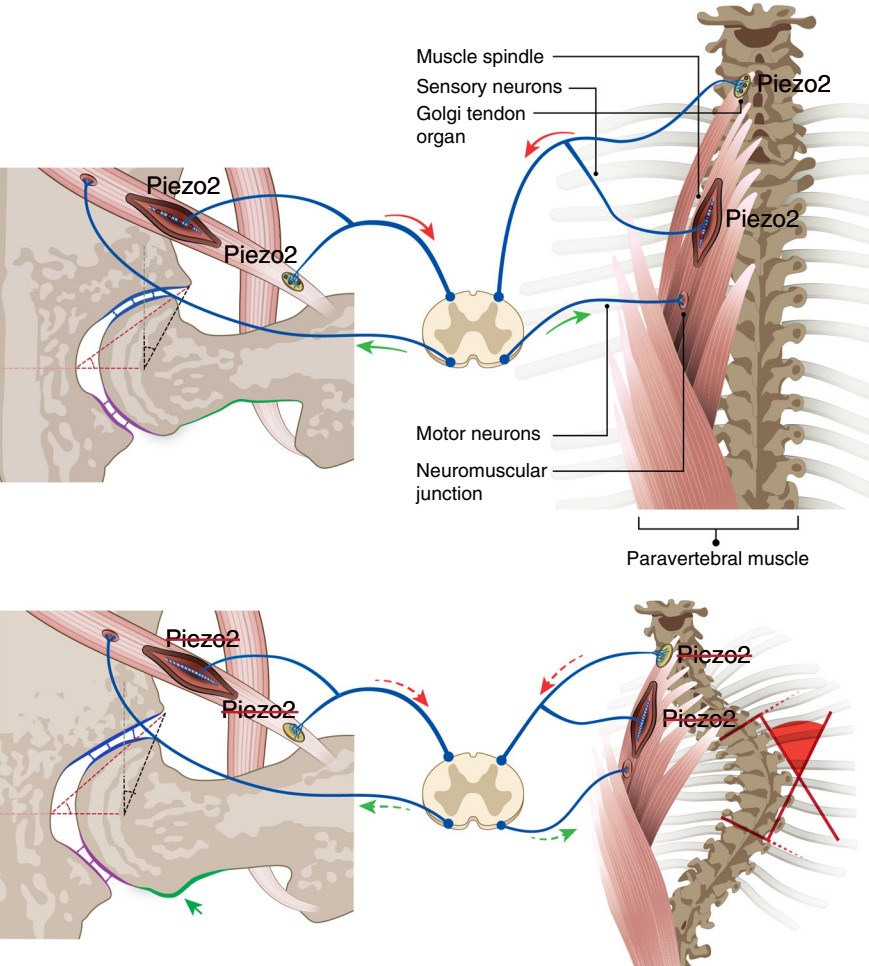

**Fig. 10 Loss of *Piezo2* in proprioceptive neuron results in spine malalignment and hip dysplasia.** In the suggested model, Piezo2 is part of the mechanism whereby the proprioceptive system regulates the activity of skeletal muscles. In the presence of Piezo2 (top), these signals maintain proper spine alignment and hip joint morphogenesis. However, the loss of Piezo2 in proprioceptive neurons (bottom) disrupts this signaling, which results in spinal malalignment and aberrant hip joint morphology.

movement that are translated into specific molecular signals are needed to regulate the alignment of three different bones to form a joint. Regarding the process that permits late changes in the hip joint, its necessity can be explained by the need of the hip joint to coordinate with changes in pelvic inclination, which result from changes in spine curvatures that occur during early life[62].

Another evidence for the significance of the proprioceptive system in joint biology is the development of a femoral cam in mice with abnormal proprioception. A possible mechanism to explain the formation of ectopic articular cartilage on the femoral neck is that the existing articular cartilage expands over the femoral neck and covers it. Alternatively, it is possible that the cells of the neck differentiate into articular cells. Since we know very little on the mechanism that drives the differentiation of articular cells, deciding between these options would be important. Either way, specific motion patterns may act to restrict articular differentiation in the neck or the abnormal cartilage expansion. From a human orthopedic point of view, the combination of hip dysplasia, manifested by shallow acetabulum, with the existence of femoral cam is interesting. In humans, femoral cam frequently coincides with extension of the upper acetabular rim (acetabular pincer) and normal CEA. Although pincer and cam can develop separately in humans, the common appearance of both suggests that the femoral cam is the result of repetitive microtrauma, possibly caused by the pincer knocking over the

femoral neck. Our finding of a femoral cam in a hip with a shallow acetabulum clearly suggests that it can be formed by a microtrauma-independent mechanism.

Previously, we showed that complete blockage of proprioceptive signaling by perturbing the genesis of muscle spindles and GTOs results in spine malalignment[38]. In this work, instead of preventing the development of this system, we targeted a specific molecular component in it, namely Piezo2. This allowed us to provide a central molecular component in this physiological regulatory pathway. The establishment of Piezo2 as an important player in this regulatory mechanism provides a molecular entry point into this pathway. The observed difference in spine malalignment, namely C-shaped scoliosis exhibited by *PValb-Cre; Piezo2^{f/f}* mice, as compared with an S-shaped scoliosis in *Runx3* and *Egr3* KO mice, indicates a variance in severity among genotypes. We previously showed a difference in the severity of scoliosis between *Runx3* KO mice, where all proprioceptors are lost, and *Egr3* KO mice[38], which lack only spindles. This suggests that level of loss of proprioception correlates with the level of impairment in spine alignment. These results may also suggest that PIEZO2 is not the sole mechanosensitive ion channel that participates in this mechanism, and that there may be a backup for its activity. Indeed, a cooperative effect of Piezo1 and Piezo2 was previously demonstrated to be required for baroception[23]. Other possibilities for backup mechanisms are members of the

major channel families degenerin/epithelial Na channels (DEG/ENaC), transient receptor potential (TRP), and the acid sensitive ion channels (ASICs)[14,21].

In this work, we aimed to trace the human phenotype caused by mutation in the *PIEZO2* gene[26]. However, while the loss of *Piezo2* in proprioceptive neurons successfully recapitulated spine malalignment and hip dysplasia, the cause for other phenotypes like hand contracture remains unknown. One explanation for the absence of some elements from the whole phenotype is the possible function of Piezo2 in tissues other than skeletal or neuronal. An alternative explanation is an accumulative effect caused by loss of *Piezo2* expression in several tissues at once.

Finally, while we focused in this work on the nonautonomous role of Piezo2 in the development of scoliosis and hip dysplasia, based on previous studies that identify the expression of *Piezo2* in articular chondrocytes[23], we do not rule out a possible cell-autonomous role of Piezo2 in skeletal biology. Our finding of reduced bone mineral density upon depletion of *Piezo2* from mesenchymal cells raises the question of the possible role of Piezo2 in bone mineralization. The use of a wide-range Cre driver as *Prx1* prevented the identification of the specific mesenchyme-derived tissues that contribute to the reduction in mineral density. However, an interesting hypothesis is that loss of *Piezo2* results in failed response of osteoblasts to mechanical stress. Further investigation is required to put this hypothesis to the test.

In summary, we identify a new nonautonomous role for Piezo2 in skeletal biology using mouse genetics. The phenotype we observed bears resemblance to that of humans with mutations in *PIEZO2*. This implies that our results may shed light on the mechanism underlying the human phenotype and provide a new animal model for studying these human pathologies. Moreover, our findings widen the scope of the regulatory roles of the proprioceptive system in skeleton biology and, as a consequence, its involvement in the etiology of various orthopedic pathologies. Finally, our results raise the possibility that movement may contribute to the treatment of pathologies that are now treated by mechanical fixation and motion restriction.

## Methods

**Experimental model and subject details.** All experiments involving mice were approved by the Institutional Animal Care and Use Committee (IACUC) of the Weizmann Institute. The generation of *Runx3*-null (KO; ICR background)[45], *Egr3*-null (KO; C57BL/6 background)[49], loxP-flanked (floxed) *Runx3* (*Runx3loxP/loxP*)[63], *Col1a1-Cre*[35], *Col2a1-Cre*[36], *Wnt1-Cre*[64], *Pvalb-Cre*[65], and floxed *Piezo2* (*Piezo2-loxP/loxP*) mice[17] have been described previously. *Col1a1-Runx3*, *Col2a1-Runx3*, and *Wnt1-Runx3* cKO mutants were generated by crossing males bearing the relevant Cre and a single *Runx3*-floxed allele (loxP/+) with a female homozygous for the *Runx3* loxP allele. *Pvalb-Piezo2*, *Col1a1-Piezo2*, *Col2a1-Piezo2*, and *Prx1-Piezo2* cKO mutants were generated by crossing males bearing the relevant Cre and a single *Piezo2*-floxed allele (loxP/+) with a female homozygous for the *Piezo2* loxP allele. In each strain, animals lacking Cre (loxP/loxP or loxP/+) served as a control. With the exception of *Runx3* (ICR) and *Egr3* (Bl/6) null mutants, all other strains were of mixed background.

**Histology.** For hematoxylin and eosin (H&E) and Safranin O staining, mice were fixed overnight in 4% PFA-PBS, decalcified in Rapid decalcifier solution (Kaltek, LOT 1738) with daily replacement of solution over 10 days, dehydrated to 70% ethanol, embedded in paraffin and sectioned at a thickness of 7 μm. Staining was performed following standard protocols.

**In situ hybridization (ISH).** Mouse were sacrificed at postnatal day (P) 10, dissected and fixed in 4% paraformaldehyde (PFA)/PBS at 4 °C overnight. After fixation, tissues were dehydrated to 70% EtOH and embedded in paraffin. The embedded tissues were cut to 7-μm thick sections and mounted onto slides. Fluorescent ISH was performed using digoxigenin (DIG)-labeled RNA probes for *Col1a1*, *Col2a1*, *Col10a1*, *Ihh*, and *Scx* (Supplementary Table 1)[66] (Shwartz and Zelzer, 2014). Probe was detected using anti-DIG-POD (1:300, 11207733910, Roche), followed by Cy3- tyramide-labeled fluorescent dyes, according to the instructions of the TSA Plus Fluorescent Systems Kit (PerkinElmer). Finally, slides were counterstained using DAPI (1:1000, D9542, Millipore Sigma).

**Ex vivo micro-computed tomography.** For ex vivo micro-CT imaging, tissue was fixed overnight in 4% PFA-PBS and dehydrated to 100% ethanol. Scans were performed with the samples immersed in ethanol using an Xradia MicroXCT-400 scanner. The source was set at 40 KV and 200 mA. We took 1500 projections over 180° with final voxel size ranging from 4 to 9 μm. Volume reconstruction was done with a proprietary Zeiss software, XMReconstructor 8.2.2720. Reconstructed datasets of each animal were merged using MATLAB software, version R2017a and reoriented using Microview software, version 2.5.0 (Parallax Innovations). 3D modeling and bone morphometric analysis were performed using Avizo software, version 9.4 (Thermo Fisher Scientific).

**In vivo micro-computed tomography.** Prior to micro-CT scanning, mice were anesthetized by isoflurane using inhalation chamber, with maintenance by inhalation mask during the scan. In vivo scans were performed using TomoScope 30 S Duo scanner (CT Imaging, Germany) equipped with two source-detector systems. The operation voltage of both tubes was 40 kV. The integration time was 90 ms (360 rotations) for 3-cm-long segments and axial images were obtained at an isotropic resolution of 80 m. Due to the length limit, imaging was occasionally performed in two overlapping parts that were then merged into one dataset representing the entire region of interest. The radiation dose range was 2.1-4.2 Gy. All micro-CT scans were reconstructed with a filtered back-projection algorithm using scanner software. Then, the reconstructed datasets of each animal were merged using Image J software, version 1.52a (imagej.net). 3D volume rendering images were produced using Amira software, version 5.2.2 (Thermo Fisher Scientific).

**Measurements of spinal deformity.** To measure the severity of major spinal curves, we calculated the previously described Cobb angle[37] (Cobb, 1948). For that, we first performed in vivo CT scan of the entire spine. We then identified the vertebrae that were the most side-tilted rostrally and caudally in the coronal plane, termed end-vertebrae. The angle between a line parallel to the superior endplate of the rostral end-vertebra and a line parallel to the inferior endplate of the caudal end-vertebra was measured. Positive values represent right-sided curves and vice versa. For the measurement of kyphosis, we measured similar Cobb angles between lines parallel to the superior and inferior end-vertebrae in the sagittal plane.

**Statistical analyses.** Absolute scoliosis and kyphosis values were tested using a linear mixed effects model, with genotype and time as fixed effects, and a random intercept for each mouse. Differences between genotypes were also tested separately for each time point using Mann–Whitney tests. Variances were compared between genotypes per time point using *F*-tests. Indices of hip deformity (see below) were compared between genotypes using Welch's two-sample *t*-tests. Mann–Whitney tests were used in case of nonsignificant *t*-test results due to high variance of the sample.

**Morphometric analysis.** To assess the three-dimensional morphology of vertebrae, CT data were analyzed using a predefined index for each plane. We measured the ratios between right and left and between the anterior and posterior vertebral body heights in the coronal and sagittal planes, respectively[41]. In the axial plane, we identified the center of the posterior lamina of each vertebra and the center of the superior facet joint[42]. Next, we measured the angle between the center points at each side and calculated the ratio between them.

**Image registration.** For qualitative and visual morphological comparison between vertebrae, we matched the surfaces of pairs of vertebrae (control vs *Pvalb-Cre; Piezo2f/f* mutant or control vs control) by image registration. Because mutant vertebrae exhibited substantial reduction in size, we performed isotropic scaling of the target image, resulting in 7 degrees of freedom (DOF; 3 translation, 3 rotation, and 1 scale). To that end, we first generated for each image a triangle mesh over the iso-surface corresponding to the external margins of the vertebra, based on an iso-value manually set according to the characteristic density of the imaged vertebra. Then, the orientation and spatial position of each surface was manually standardized, thereby generating an initial approximation of the anatomical match. To fine-tune the match between surfaces, we used the Surface Rigid Registration (srreg) module of the freely available Image Registration Toolkit (IRTK)[67–69] with a point locator. To allow isotropic scaling during registration, we implemented an iterative algorithm to identify the optimal spatial transformation by multiplying the coordinates of the vertices of the target surface with a scale factor, followed by fine-tuning using the ssreg module. The scale factor was manually set to range from 0.6 to 1, with 0.005 increments. The registration that had reached the minimal score was considered the final transformation.

**Immunofluorescence staining.** For immunofluorescence staining, 7-μm thick paraffin sections of embryo vertebra were deparaffinized and rehydrated in water. Antigen retrieval for GFP and F4/80 antibodies was performed in 10 mM sodium citrate buffer (pH 6.0) heated at 80 °C for 15 min. Then, sections were washed twice in PBS and endogenous peroxidase was quenched using 3% $H_2O_2$ in PBS. Non-specific binding was blocked using 7% horse serum and 1% BSA dissolved in PBST

for 1 h. Then, sections were incubated with rat Anti-F4/80 antibody (1:50, ab6640, Abcam) overnight at room temperature. The next day, sections were washed twice in PBST and incubated with biotin anti-rat (1:100 Jackson laboratories) for 1 h at room temperature. Then, after two washes of PBST, slides were incubated with streptavidin-Cy3 (1:100, 615-222-214, Jackson ImmunoResearch), and Cy3-conjugated donkey anti-rat (1:100, 712-165-153, Jackson ImmunoResearch) for 1 h at room temperature. Occasionally, slides were counterstained using DAPI. Then, slides were mounted with Immu-mount aqueous-based mounting medium (Thermo Fisher Scientific).

**Measurements of hip deformity.** To assess the severity of hip dysplasia, we measured the previously described lateral central edge angle, which will be referred to in the following as central edge angle (CEA)[32] and acetabular index[33], as well as the newly introduced congruency index. For that, we first performed ex vivo CT of the hip joint. We then aligned the scan using Microview software to face the true coronal plane of the hip. The CEA was measured between two lines drawn from the center of rotation at the femoral head: a vertical upward line and a second line to the acetabular sourcil. The acetabular index was measured between a horizontal line connecting both acetabular centers (Hilgenreiner line) and a line extending from the Hilgenreiner line deep in the acetabulum into the sourcil. The acetabular index was measured separately for the upper and lower acetabular roof. It was calculated using multiple measurements of the distance along the joint line divided by the minimal value out of all these measurements. A value of 1 indicates perfect congruency, whereas incongruence results in higher values.

**Reporting summary.** Further information on research design is available in the Nature Research Reporting Summary linked to this article.

## Data availability

The source data underlying Figs. 1e, 2b, d, 3b, 4b–d, 5a–d, 6e, 7c, 8c and 9b and Supplementary Figs. 1e and 4c are provided as a Source Data file. Other data that support the findings of this study are available from the corresponding author upon reasonable request. Source data are provided with this paper.

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

## Acknowledgements
We thank Ms. Hanna Vega from the Design Section of the Information Systems Division at the Weizmann Institute for her help with the model figure, Mr. Nitzan Konstantin for assisting in writing the manuscript, and all other member of the Zelzer lab for support and advice. This study was funded by grants from the Israel Ministry of Science, Technology and Space (IMOS-Tashtiot grant #713272), the Israel Science Foundation Legacy Heritage. Biomedical Science Partnership (grant #2147/17), the Yves Cotrel Foundation, the Estate of Mr. and Mrs. van Adelsbergen, the Julie and Eric Borman Family Research Fund, and the David and Fela Shapell Family Center for Genetic Disorders.

## Author contributions
E.A. designed the study, conducted experiments, analyzed the results, and wrote the manuscript; R.B. analyzed the scoliotic phenotype and contributed to writing the manuscript; L.H.Y. contributed to mouse breeding; S.K. and R.C.V. conducted in situ hybridization and immunohistochemistry; I.E.B conducted in vivo mouse imaging; V.B. conducted ex vivo mouse imaging; R.R. conducted statistical analyses; E.A. and G.A. conceptualized the project; E.Z. designed and supervised the study, analyzed the results, and wrote the manuscript.

## Competing interests
The authors declare no competing interests.
