## [Peer Review File · Nature Communications]

Reviewers' Comments:

Reviewer #1:

Remarks to the Author:

Mutations in PIEZO2, a gene that encodes a mechanically activated cation channel, are associated with skeletal abnormalities in humans. Assaraf et al. used mouse genetics to study roles of Piezo2 in the proprioceptive systems in regulating skeletal development and morphology. They showed that Piezo2 expression in bone cells (chondrocytes or osteogenic lineages) or mesenchyme-derived tissues (i.e. joint components) does not cause defects in skeleton and joint morphology. Remarkably, they observed skeletal deficits such as spinal mal-alignment, hip dysplasia, and joint deformity in mice lacking Piezo2 specifically in proprioceptive neurons and these phenotypes recapitulate human patient conditions. Further, they genetically disrupted proprioceptive circuitry and muscle spindles by Runx3 (peripheral neuron specific) and Egr3 mutants and observed hip dysplasia phenotypes. This confirms their hypothesis of this and previous work that proprioceptive system is important in skeletal development.

In this manuscript, Piezo2 loss-of-function genetic analyses are elegant and convincing. These data not only provides more support for their previous work (Bletcher et al., 2017) that proprioceptive system is an essential component in skeletal integrity but also established Piezo2 as a key regulator of this mechanism. However, when they reported phenotypes of Runx3 and Egr3 mutants, they did not comprehensively describe skeletal abnormalities, for example, whether peripheral neuron specific loss of Runx3 leads to scoliosis and/or kyphosis? They have reported some of these phenotypes in Runx3 and Egr3 mutants in previous work (Bletcher et al., 2017), but it will still be necessary to compare them to phenotypes observed in Piezo2 proprioceptive neuron specific KO. Minor point: 1) in Introduction part, they mentioned " conformational change of the distal blades slightly opens the central pore, allowing cation-selective permeation^{3, 4, 5} " But there is no evidence to support this statement. 2) scale bars are missing in Figure 1 and 3. 3) Figure 4: Graph should be labeled as "D"

Reviewer #2:

Remarks to the Author:

The current manuscript describes that loss of Piezo2 in proprioceptive neurons, but not in osteoblasts or chondrocytes led to scoliosis and hip dysplasia, suggesting that the Piezo2 in proprioceptive neurons can regulate skeleton in a non-autonomous way. The role of proprioception system in the regulation of hip joint morphogenesis was further confirmed by mice mutant for Runx3 and Egr3, which have similar phenotype with Piezo2 mutant mice. The same group published a paper in Developmental Cell (2017, ref 1) entitled 'the The Proprioceptive System Masterminds Spinal Alignment: Insight into the Mechanism of Scoliosis', uncovering the central role of proprioception system in maintaining spinal alignment. The current manuscript confirms and expands their previous concept by at least two ways: one is that the skeleton phenotype has been expanded to the hip joint morphogenesis; another is that Piezo2 deletion in proprioceptive neurons regulates skeletal integrity. However, considering that Piezo2 has been well demonstrated as the principal mechanotransduction channel for proprioception and PValb-Cre, Piezo2 cKO mice have shown severely abnormal limb positions in previous publications (ref 2, 3), further mechanism study by which Piezo2 regulates skeleton integrity is needed to guarantee the publication in Nature Communications.

1- The paper from the same group has demonstrated that Runx3 or Egr3 mutant mice lead to scoliosis. I would suggest the author weaken the point about Runx3 or Egr3 mutant mice in Abstract and focus more on the role of Piezo2 in proprioceptive neurons regulating skeleton integrity.

2- The author has emphasized that vertebral morphology might not play a causative role in the scoliosis. Does Piezo2 deletion in PValb+ cells lead to the damage to neuron, muscle and

intervertebral discs? This may contribute to scoliosis.

3- The authors did skeleton phenotype analysis on spine and hip joint. It is hard to conclude that 'loss of piezo2 in chondrogenic or osteogenic lineages don't need to a skeletal phenotype'. Does Col1-cre or Col2 Cre Piezo2 CKO mice have limb development phenotype? Bone density phenotype?

4- The proprioceptive system include two types of peripheral sensors, muscle spindles and the GTO, and both receptors are acquired for the formation of normal hip joint. It is worthy to check Piezo2 expression in the two types of sensors to further clarify the Piezo2 functions in muscle spindles or the GTO, or both.

5- Does the alternation of hip morphology cause the joint wearing or inflammation in the cartilage or ligament?

6- To link the different mice models into a whole story, the author may check whether Runx3 and/or Egr3 regulate the expression of Piezo2.

7- In the Abstract and Discussion, the authors highlight Piezo2 as the first component or molecular understanding of how the proprioceptive system regulates the skeleton. I would suggest the author weaken the point as Runx3 and Egr3 has been reported in their previous paper.

Reference:

1. Blecher, R., et al., The Proprioceptive System Masterminds Spinal Alignment: Insight into the Mechanism of Scoliosis. *Dev Cell*, 2017. 42(4): p. 388-399 e3.
2. Woo, S.H., et al., Piezo2 is the principal mechanotransduction channel for proprioception. *Nat Neurosci*, 2015. 18(12): p. 1756-62.
3. Florez-Paz, D., et al., A critical role for Piezo2 channels in the mechanotransduction of mouse proprioceptive neurons. *Sci Rep*, 2016. 6: p. 25923.

Dear reviewers,

Please find attached the revised version of our manuscript “**Piezo2 Expressed in Proprioceptive Neurons is Essential for Skeletal Integrity**”. We thank you for your constructive criticism, which helped us to improve the manuscript considerably. In the following are point-by-point responses to your comments. We hope that you find the revised manuscript to be ready for publication in *Nature Communications*.

Comments and responses

Reviewer 1

Mutations in PIEZO2, a gene that encodes a mechanically activated cation channel, are associated with skeletal abnormalities in humans. Assaraf et al. used mouse genetics to study roles of Piezo2 in the proprioceptive systems in regulating skeletal development and morphology. They showed that Piezo2 expression in bone cells (chondrocytes or osteogenic lineages) or mesenchyme-derived tissues (i.e. joint components) does not cause defects in skeleton and joint morphology. Remarkably, they observed skeletal deficits such as spinal mal-alignment, hip dysplasia, and joint deformity in mice lacking Piezo2 specifically in proprioceptive neurons and these phenotypes recapitulate human patient conditions. Further, they genetically disrupted proprioceptive circuitry and muscle spindles by Runx3 (peripheral neuron specific) and Egr3 mutants and observed hip dysplasia phenotypes. This confirms their hypothesis of this and previous work that proprioceptive system is important in skeletal development.

In this manuscript, Piezo2 loss-of-function genetic analyses are elegant and convincing. These data not only provides more support for their previous work (Bletcher et al., 2017) that proprioceptive system is an essential component in skeletal integrity but also established Piezo2 as a key regulator of this mechanism. However, when they reported phenotypes of Runx3 and Egr3 mutants, they did not comprehensively describe skeletal abnormalities, for example, whether peripheral neuron specific loss of Runx3 leads to scoliosis and/or kyphosis? They have reported some of these phenotypes in Runx3 and Egr3 mutants in previous work (Bletcher et al., 2017), but it will still be necessary to compare them to phenotypes observed in Piezo2 proprioceptive neuron specific KO.

We thank the reviewer for the useful comment. Following the suggestion, we added a new figure (Fig 5) describing the kyphosis phenotypes in both *Egr3* KO and *Runx3* KO mice and we now compare between these findings and kyphosis in *Piezo2* cKO mice. Moreover, we added box plot graphs to Fig 5 comparing scoliosis and kyphosis phenotypes among the three mouse lines.

Minor point:

1) in Introduction part, they mentioned “ conformational change of the distal blades slightly opens the central pore, allowing cation-selective permeation^{3, 4, 5} ” But there is no evidence to support this statement.

The sentence has been deleted.

2) scale bars are missing in Figure 1 and 3.

Scale bars have been added.

3) Figure 4: Graph should be labeled as “D”

Graph has been labeled as suggested.

Reviewer 2

The current manuscript describes that loss of Piezo2 in proprioceptive neurons, but not in osteoblasts or chondrocytes led to scoliosis and hip dysplasia, suggesting that the Piezo2 in proprioceptive neurons can regulate skeleton in a non-autonomous way. The role of proprioception system in the regulation of hip joint morphogenesis was further confirmed by mice mutant for Runx3 and Egr3, which have similar phenotype with Piezo2 mutant mice. The same group published a paper in *Developmental Cell* (2017, ref 1) entitled ‘the The Proprioceptive System Masterminds Spinal Alignment: Insight into the Mechanism of Scoliosis’, uncovering the central role of proprioception system in maintaining spinal alignment. The current manuscript confirms and expands their previous concept by at least two ways: one is that the skeleton phenotype has been expanded to the hip joint morphogenesis; another is that Piezo2 deletion in proprioceptive neurons regulates skeletal integrity. However, considering that Piezo2 has been well demonstrated as the principal mechanotransduction channel for proprioception and PValb-Cre, Piezo2 cKO mice have shown severely abnormal limb positions in previous publications (ref 2, 3), further mechanism study by which Piezo2 regulates skeleton integrity is needed to guarantee the publication in *Nature Communications*.

1- The paper from the same group has demonstrated that Runx3 or Egr3 mutant mice lead to scoliosis. I would suggest the author weaken the point about Runx3 or Egr3 mutant mice in Abstract and focus more on the role of Piezo2 in proprioceptive neurons regulating skeleton integrity.

We agree with the reviewer that the main focus should be on the role of Piezo2 in proprioceptive neurons regulating skeleton integrity. Indeed, this is the main point of the Abstract. Runx3 and Egr3 are mentioned only in the context of the novel joint phenotype, which, to our knowledge, has never been described for these genes, and in support of the function of Piezo2 in skeletal regulation.

2- The author has emphasized that vertebral morphology might not play a causative role in the scoliosis. Does Piezo2 deletion in PValb+ cells lead to the damage to neuron, muscle and intervertebral discs? This may contribute to scoliosis.

Indeed, neuronal deletion of *Piezo2* could damage the neurons, muscles or intervertebral disks, which could potentially contribute to the phenotype. As we do not specialize in analysis of neuron or muscle tissues, we could not negate this possibility altogether. However, to address this issue, we dissected the vertebral body and surrounding tissues using both histological and gene expression analyses. As seen in the newly added Supplementary Figure S1, the results do not support the possibility that morphological or developmental changes to these tissues contribute to the phenotype.

3- The authors did skeleton phenotype analysis on spine and hip joint. It is hard to conclude that 'loss of *piezo2* in chondrogenic or osteogenic lineages don't need to a skeletal phenotype'. Does *Col1-cre* or *Col2 Cre Piezo2* CKO mice have limb development phenotype? Bone density phenotype?

Following this comment, we analyzed the possible involvement of *Piezo2* in embryonic and postnatal skeletal development. For that, we performed histological, gene expression and micro-CT analyses on *Prx1-Cre;Piezo2^{f/f}* cKO mice and control littermates. Based on the new data shown in the newly added Figure 1, we concluded that *Piezo2* does not play a major role for in skeletogenesis. However, postnatal micro-CT images did reveal a mild reduction in bone mineral density in mice lacking *Piezo2* in limb skeletal progenitors, a result that could not account for the observed phenotype.

4- The proprioceptive system include two types of peripheral sensors, muscle spindles and the GTO, and both receptors are acquired for the formation of normal hip joint. It is worthy to check *Piezo2* expression in the two types of sensors to further clarify the *Piezo2* functions in muscle spindles or the GTO, or both.

We agree with the reviewer that the question of *Piezo2* expression in GTOs and spindles is very interesting. However, we believe that this question have already been addressed sufficiently in the paper by Woo et al. (2015), who showed that *Piezo2* is indeed expressed in these receptors. We now cite these findings in the Discussion.

5- Does the alternation of hip morphology cause the joint wearing or inflammation in the cartilage or ligament?

To address this question, we analyzed histological joint sections for signs of articular cartilage degradation using Safranin O staining and ISH for the articular cartilage marker lubricin. Results showed no sign for wear.

Moreover, immunofluorescence staining for the macrophage marker F4-80 revealed no signs of an inflammatory process around the joint line. These data are shown in the newly added Supplementary Figure S2.

6- To link the different mice models into a whole story, the author may check whether *Runx3* and/or *Egr3* regulate the expression of *Piezo2*.

Following this suggestion, we stained DRG sections from P60 *Egr3* KO mice and control littermates for PIEZO2 protein. As seen in the figure below, no difference in expression was found, suggesting that EGR3 does not regulate *Piezo2* expression. With regard to Runx3, since in *Runx3* KO mice proprioceptive TrkC neurons fail to survive (Levanon et al., 2002), it is impossible to examine *Piezo2* expression by these neurons.

Expression of *Egr3* is dispensable for PIEZO2 expression in DRG neurons. Immunofluorescence staining for PIEZO2 applied on histological sections of the DRG from *Egr3* KO mice (right) and control littermates at P60 shows similar expression.

7- In the Abstract and Discussion, the authors highlight Piezo2 as the first component or molecular understanding of how the proprioceptive system regulates the skeleton. I would suggest the author weaken the point as Runx3 and Egr3 has been reported in their previous paper.

We thank the reviewer for the comment, and we have toned down our statements accordingly.

References

- Levanon, D. *et al.* The Runx3 transcription factor regulates development and survival of TrkC dorsal root ganglia neurons. *EMBO J.* **21**, 3454–63 (2002).
- Woo, S. H. *et al.* Piezo2 is the principal mechanotransduction channel for proprioception. *Nat. Neurosci.* **18**, 1756–1762 (2015).

Reviewers' Comments:

Reviewer #1:

Remarks to the Author:

The authors used elegant mouse genetics to study the role of Piezo2, a mechanically activated ion channel, in proprioceptive neurons to regulate skeletal development and morphology. This work not only confirms their previous hypothesis that proprioceptive system is an essential component in skeletal integrity but also determines that PIEZO2 is a key regulator of this interaction.

In response to our comments the authors compared skeletal abnormalities between proprioceptive neuron specific PIEZO2 KO mice with Runx3 and Egr3 KO mice that have disrupted proprioception circuitry as well as muscle spindles. They claim that Piezo2 KO mutants have similar pattern of skeletal mal-alignment as Runx3 or Egr3 mutants. However, the authors did not show statistical significance when comparing above genotypes. Indeed, the variation is quite noticeable in Fig 5C that even wild type controls for Egr3 KO and Piezo2 KO seem to show a difference. For readers to better evaluate the true effects, I suggest that the authors to provide detailed statistics. In fact, they found Piezo2KO mice have C-shape scoliosis whereas Runx3 and Egr3 KO showed S-shape scoliosis. It seems to suggest a less severe skeletal defect in Piezo2KO. We suggest they add some discussion on this point.

They have corrected the minor points we raised in the previous review comment.

Reviewer #2:

Remarks to the Author:

The authors have answered all of my questions and the paper has been significantly improved. The findings in the paper help to understand the contributions and the mechanisms of the regulations of proprioceptive system on skeleton system. Congratulations on this important study.

Reviewer 1 comments

The authors compared skeletal abnormalities between proprioceptive neuron specific PIEZO2 KO mice with Runx3 and Egr3 KO mice that have disrupted proprioception circuitry as well as muscle spindles. They claim that Piezo2 KO mutants have similar pattern of skeletal mal-alignment as Runx3 or Egr3 mutants. However, the authors did not show statistical significance when comparing above genotypes. Indeed, the variation is quite noticeable in Fig 5C that even wild type controls for Egr3 KO and Piezo2 KO seem to show a difference. For readers to better evaluate the true effects, I suggest that the authors to provide detailed statistics.

A- As suggested by the reviewer, we have added to the Result section detailed statistics (Levene's test result) on the variation in skeletal malalignment among the different mouse lines.

In fact, they found Piezo2KO mice have C-shape scoliosis whereas Runx3 and Egr3 KO showed S-shape scoliosis. It seems to suggest a less severe skeletal defect in Piezo2KO. We suggest they add some discussion on this point.

A- As suggested by the reviewer, we have added discussion on the variation in the pattern of scoliosis between Piezo2 KO mice and Runx3 and Egr3 KO mice.